# Integrin-Linked Kinase (ILK) Promotes Mitochondrial Dysfunction by Decreasing CPT1A Expression in a Folic Acid-Based Model of Kidney Disease

**DOI:** 10.3390/ijms26051861

**Published:** 2025-02-21

**Authors:** Mariano de la Serna-Soto, Laura Calleros, María Martos-Elvira, Ariadna Moreno-Piedra, Sergio García-Villoria, Mercedes Griera, Elena Alcalde-Estévez, Ana Asenjo-Bueno, Diego Rodríguez-Puyol, Sergio de Frutos, María Piedad Ruiz-Torres

**Affiliations:** 1Department of Systems Biology, Universidad de Alcalá, Instituto Ramon y Cajal de Investigación Sanitaria, RICORS 2040, Fundación Renal Iñigo Álvarez de Toledo, INNOREN-CM, Alcalá de Henares, 28871 Madrid, Spain; mariano.serna@uah.es (M.d.l.S.-S.); laura.calleros@uah.es (L.C.); maria.martos@uah.es (M.M.-E.); ariadna.moreno@uah.es (A.M.-P.); sergio.garciavillori@uah.es (S.G.-V.); elena.alcaldee@uah.es (E.A.-E.); ana.asenjo@uah.es (A.A.-B.); mpiedad.ruiz@uah.es (M.P.R.-T.); 2Graphenano Medical Care S.L., Alcalá de Henares, 28871 Madrid, Spain; mercedesgriera@graphenano.com; 3Department of Medicine, Universidad de Alcalá, Nephrology Service at Hospital Príncipe de Asturias, Instituto Ramon y Cajal de Investigación Sanitaria, RICORS 2040, Fundación Renal Iñigo Álvarez de Toledo, INNOREN-CM, Alcalá de Henares, 28871 Madrid, Spain; diego.rodriguez@uah.es

**Keywords:** ILK, kidney disease, folic acid, HK2, AKI-to-CKD transition, fibrosis, autophagy, mitochondria, oxidative phosphorylation, GSK3β, C/EBPβ, CPT1A

## Abstract

Integrin-linked kinase (ILK) is a key scaffolding protein between extracellular matrix protein and the cytoskeleton and has been implicated previously in the pathogenesis of renal damage. However, its involvement in renal mitochondrial dysfunction remains to be elucidated. We studied the role of ILK and its downstream regulations in renal damage and mitochondria function both in vivo and vitro, using a folic acid (FA)-induced kidney disease model. Wild type (WT) and ILK conditional-knockdown (cKD-ILK) mice were injected with a single intraperitoneal dose of FA and studied after 15 days of chronic renal damage progression. Human Kidney tubular epithelial cells (HK2) were transfected with specific siRNAs targeting ILK, glycogen synthase kinase 3-β (GSK3β), or CCAAT/enhancer binding protein-β (C/EBPβ). The expressions and activities of renal ILK, GSK3β, C/EBPβ, mitochondrial oxidative phosphorylation enzymes, and mitochondrial membrane potential were assessed. Additionally, the expression of markers for fibrosis fibronectin (FN) and collagen 1 (COL1A1), for autophagy p62 and cytosolic light chain 3 (LC3B) isoforms II and I, and mitochondrial homeostasis marker carnitine palmitoyl-transferase 1A (CPT1A) were evaluated using immunoblotting, RT-qPCR, immunofluorescence, or colorimetric assays. FA upregulated ILK expression, leading to the decrease of GSK3β activity, increased tubular fibrosis, and produced mitochondrial dysfunction, both in vivo and vitro. These alterations were fully or partially reversed upon ILK depletion, mitigating FA-induced renal damage. The signaling axis composed by ILK, GSK3β, and C/EBPβ regulated CPT1A transcription as the limiting factor in the FA-based impaired mitochondrial activity. We highlight ILK as a potential therapeutical target for preserving mitochondrial function in kidney injury.

## 1. Introduction

Acute kidney injury (AKI) may appear within a few hours or days following renal insult. In contrast, chronic kidney disease (CKD) appears from persistent injury, leading to interstitial accumulation of extracellular matrix (ECM), known as fibrosis, which conditions the progressive decline of both glomerular and tubular function. AKI and CKD are interconnected syndromes with rising incidences and poor outcomes [1]. Additionally, if the resolution of AKI fails, the damaged renal cells drive to AKI-to-CKD transition or CKD progression, ultimately leading to kidney fibrosis [2]. A key signaling component between ECM and cellular phenotype changes, both transcriptional and posttranscriptional, is the intracellular scaffold protein Integrin-linked kinase (ILK), which is associated with the actin cytoskeleton architecture and the phosphorylation of protein kinase B (also known as AKT) and glycogen synthase kinase 3-β (GSK3β) [3]. GSK3β is a constitutively active kinase that can be phosphorylated at serine 9, causing its inhibition by several upstream kinases, such as AKT and ILK [4]. We have previously demonstrated [5,6,7,8,9] the role of ILK in the pathogenesis of CKD and AKI in several in vitro and vivo models, a finding that has been corroborated by others [10,11,12,13,14]. Moreover, the inactivation of ILK downstream target GSK3β, either via transgenic depletion or by Serine 9 phosphorylation, correlates with exacerbated kidney damage in multiple AKI and CKD models, both in vitro and in vivo, by mechanisms related to mitochondrial metabolism and permeability, transcriptional regulation, oxidative stress and tubular epithelial cells survival [15,16]. In this sense, mitochondrial function is crucial for the kidney since it is one of the most energy-demanding organs. To meet cellular requirements, renal cells can adjust the mitochondrial content and function by modifying mitochondrial biogenesis, fusion, fission, and mitophagy of dysfunctional mitochondria. Mitochondrial dysfunction is a hallmark of renal pathology, manifested as reduced expression and/or activity of enzymes related to mitochondrial oxidative phosphorylation (OXPHOS), increased oxidative stress, electron transport chain disruption, and increased autophagy and mitophagy [17,18,19]. Among the various markers for renal mitochondrial homeostasis, the expression of the rate-limiting enzyme during fatty acid oxidation (FAO), carnitine palmitoyl-transferase 1A (CPT1A), is probably the most critical [20]. CPT1A is reduced in both AKI and CKD patients, as well as in experimental models, and its genetic deletion or pharmacological inhibition in animal models has been associated with increased tubulointerstitial ECM accumulation [21,22,23]. Conversely, tubular overexpression of CPT1A protects against CKD-dependent fibrosis in a mice model [24]. The regulation of CPT1A in renal pathology is not yet well known, but CCAAT/enhancer binding protein-β (C/EBPβ), a known substrate of active GSK3β [25,26], is modulated during kidney functionality in several models of renal injury [27,28]. Moreover, C/EBPβ could be implicated in the transcriptional regulation of renal CPT1A [29,30].

In the present study, we used a renal damage model based on folic acid (FA) [31] on transgenic mice and cultured human tubular epithelial cells HK2 with ILK depletion [7] to investigate the role of ILK and downstream intracellular mechanisms involved in mitochondrial dysfunction during AKI-to-CKD transition.

## 2. Results

### 2.1. Renal ILK Depletion Prevents Folic Acid (FA)-Induced Renal Injury on Mice

WT and conditional Knock-Down ILK (cKD-ILK) mice [6,7] were subjected to an AKI-to-CKD transition model based on the administration of a single dose of FA (named WT FA or cKD-ILK FA, respectively) or vehicle as control (named WT CT or cKD-ILK CT, respectively). After 15 days, animals were sacrificed, and samples were extracted [24,31,32].

Figure 1A,B show that WT FA mice have increased blood concentrations of creatinine and BUN compared to the WT CT, which means a decline of renal function after FA treatment, whereas cKD-ILK FA littermates have partially and significantly prevented the increase in these renal dysfunction markers. Figure 1C shows the histopathological analysis of tubulointerstitial damage in kidney sections of the animals after FA treatments. WT FA mice exhibited significant tubular dilatation, atrophy, and intratubular pus casts. All these tubular damage markers were significantly reduced in cKD-ILK FA. Table 1 compilates the analysis of each tubular parameter percentage with respect to the total number of tubules per area. Because the lack of ILK was related to the reduction of pathological features, we analyzed whether the renal expression of ILK was affected after FA insult. Figure 1D shows that WT FA has increased ILK renal expression compared to WT CT. Both cKD-ILK experimental groups, CT and FA, showed approximately half of the ILK content in kidneys when compared to their corresponding WT littermates, as we previously reported [6,7].

The presence or absence of ILK may have consequences in downstream effectors such as GSK3β because it has been previously stated that ILK can mediate in the grade of phosphorylation in serine 9, which inactivates the protein [3]. Therefore, we studied whether the presence of ILK in WT and the absence in cKD-ILK was affecting GSK3β expression and activity. Phosphorylation of GSK3β at serine 9 was increased in FA-treated WT but not in FA-treated cKD-ILK, while GSK3β total contents were not different between experimental groups (Figure 1E). These data demonstrate that FA overexpresses ILK in the kidney, and this is probably related to the renal damage because the depletion of ILK in cKD-ILK prevented it. Moreover, the data suggest that ILK mediates downstream phosphorylation of GSK3β and, therefore, its activity, which has been previously shown to have consequences on the protection of the kidney [15,16].

Interstitial fibrosis, understood as the inadequate deposition of extracellular matrix (ECM), is a consequence of tubular damage [2,33,34]. Figure 2 shows the analysis of expression of ECM proteins fibronectin (FN) and collagen type 1 (COL1A1) in the renal samples from the different experimental groups. In WT FA, the expressions of FN (Figure 2A) and COL1A1 (Figure 2B) were increased compared with WT CT. However, the increase in both proteins was prevented in cKD-ILK FA. Figure 2C shows the histopathological analysis of ECM interstitial deposition on kidney sections after FA treatments by using Sirius red staining and the analysis of red intensity per area photographed [5,6]. WT FA mice exhibited higher interstitial red staining corresponding to collagen depositions compared to WT CT, and this increase was prevented in cKD-ILK FA.

### 2.2. Mitochondrial Dysfunction and Autophagy in FA-Induced Renal Damage Is Mediated by ILK Overexpression

Mitochondrial dysfunction and unbalanced autophagy have been related to tubular damage in AKI and CKD [2,19,24]. Therefore, we analyzed whether ILK overexpression after FA treatment was affecting mitochondrial function and autophagy. Mitochondrial function was assessed by several parameters. Figure 3 shows a representative immunoblot (Figure 3A) and the individual densitometric analysis (Figure 3B–F) of the protein content of important enzymes for mitochondrial oxidative phosphorylation (OXPHOS) present in renal extracts from the different experimental groups. The enzymes studied are ATP Synthase (ATP5A1), Coenzyme Q (UQRC2), mitochondrial Cytochrome (Cyt) C oxidase subunit 1 (MT-CO1), Succinate Dehydrogenase (SDHB), and NADPH Dehydrogenase (NDUFB8). The analysis shows that the expression of all these enzymes is reduced after FA treatment. This is in accordance with previous works in similar renal damage models where the reduced expression of these enzymes was clearly related to mitochondrial dysfunction [19,24,33,34]. However, it is important to highlight that FA-induced downregulation of MT-CO1 and NDUFB8 expressions were partially prevented in cKD-ILK compared to WT FA (Figure 3D and Figure 3F, respectively).

Because the expression of renal mitochondrial-related Cyt C oxidase subunit MT-CO1 was downregulated by FA and prevented when ILK was depleted, we analyzed Cyt C oxidase activity of the isolated mitochondrial extracts from renal samples of the different experimental groups, using mitochondria-cytosol fractioning and activity measurement commercial kits as detailed in the Section 4. The results are shown in Figure 3G. Cyt C oxidase activity from renal isolated mitochondria from FA-treated animals was reduced, in accordance with the expression results shown in Figure 3D. Again, it is important to highlight that the reduced activity was partially prevented in cKD-ILK FA samples, although it did not reach statistical significance.

Renal tubular cells are primarily dependent on Fatty acid oxidation (FAO) as a source of ATP, which occurs in mitochondria and involves a repeated sequence of reactions that result in the conversion of fatty acids to acetyl-CoA that is recycled in the tricarboxylic acid (TCA) cycle through subsequent rounds. Thus, we studied the activity of the TCA cycle-limiting enzyme, citrate synthase, in the mitochondrial extracts isolated from renal samples. Figure 4A shows that renal mitochondrial citrate synthase activity from FA-treated mice was reduced, and this reduction was partially prevented in cKD-ILK FA samples. Then, we analyzed the expression of the Carnitine palmitoyl-transferase 1 A (CPT1A). The reduced activity on the mitochondrial TCA cycle could be caused by the downregulation or malfunction of CPT1A, which is a crucial enzyme in fatty acid metabolism, as it facilitates the transport of long-chain fatty acids across the outer mitochondrial membrane. Dysfunction or downregulation of CPT1 can negatively affect FAO, which in turn can reduce the activity of the TCA cycle in the mitochondria [24].

Figure 4B shows the decreased expression of CPT1A in WT FA renal samples. Interestingly, deletion of ILK in cKD-ILK CT induced a strong increase in CPT1A expression, and this upregulation prevented the FA-induced decrease of CPT1A in cKD-ILK FA significantly. Reduced levels of CPT1A protein have been associated with kidney damage in AKI and CKD patients and experimental models [20,24]. This group of results suggests that FA treatment induced mitochondrial dysfunction in the kidney through ILK overexpression by reducing some enzymes involved in fatty acid metabolism and OXPHOS activity.

Finally, we analyzed whether FA injections promoted changes in autophagy in renal samples through ILK overexpression. Autophagy was studied in renal extracts by the ratio between cytosolic light chain 3 isoforms II and I (LC3-II/LC3I) and by the expression of the ubiquitin-associated p62 [17,18]. Figure 5 shows the expression levels of these autophagy markers in the renal samples. FA decreased the expression of p62 and increased the LC3BII/LC3BI ratio (Figure 5A and Figure 5B, respectively), suggesting an increase in autophagy. Again, the increase in autophagy was partially prevented in cKD-ILK FA samples, revealing an important role for ILK in regulating autophagic flux as a part of the pathological effect of FA.

### 2.3. ILK Overexpression Promotes Fibrosis and Mitochondrial Dysfunction in an FA-Based Damage Model on Cultured HK2 Cells

To further analyze the implication of ILK during FA-based tubular damage, an in vitro approach based on the direct exposition to FA in cultured tubular cell line HK2 was designed. ILK upregulation observed in WT FA kidneys was also confirmed in the FA-treated HK2 model. Figure 6A shows ILK increased expression after 24 of FA treatment on HK2 in a dose-dependent manner, being the most effective dose 10 mM FA.

Then, to confirm that ILK transgenic depletion prevents the deleterious effects of FA in vitro, ILK expression was silenced in cultured HK2 using specific siRNAs (siILK). The depletion was approximately 50% in protein and mRNA levels (Figure 6B and Figure 6C, respectively), in accordance with other siILK transfected cultured cells used in our previous works [6,7]. The depletion was maintained even when FA was added to the cells, significantly preventing the potential induction of more ILK expression. This figure confirms that siILK-transfected HK2 cells can be used as an in vitro model for FA treatment, keeping similar behavior to cKD-ILK kidneys.

Following the analysis performed in the renal samples from previous figures, we analyzed whether ILK overexpression in FA-treated HK2 was involved in fibrosis and mitochondrial dysfunction [13,14]. Figure 7A,B shows the analysis of fibronectin (FN) content, both protein and mRNA, respectively. Figure 7C,D show the analysis of collagen I (COL1A1) content, both protein and mRNA, respectively. The expressions of these fibrosis markers were all increased in FA-treated cells, while siILK-transfected cells prevented these upregulations.

As in the animal model, we studied the mitochondrial activity of Cyt C oxidase and citrate synthase in HK2 cells. Figure 8A,B show that both enzymes have reduced activity after FA treatment while the reductions were partially prevented when ILK was depleted in FA-treated siILK HK2 cells. Figure 8C shows that CPT1A protein expression was downregulated in FA-treated HK2, in accordance with the in vivo model.

It is important to notice that control siILK-transfected cells (siILK CT) strongly increased their CPT1A expression, exceeding the values of the basal CT cells. This upregulation prior to being affected by FA was responsible for preventing the further FA-induced decrease of CPT1A, as can be observed in siILK FA cells. Mitochondrial activity can also be analyzed by determining the mitochondrial membrane potential. Figure 8D shows the state of the mitochondrial membrane potential in the experimental groups of HK2, as shown by the analysis of internalized fluorescent, which proves TMRM [35]. FA cells have a significant decrease in TMRM intracellular staining, which means reduced membrane potential, whereas this decrease was importantly prevented in siILK FA cells, even exceeding the values of CT cells. In summary, FA-induced fibrosis and mitochondrial dysfunction in renal tubular cells HK2 are caused by the overexpression of ILK in a similar way to the in vivo model.

### 2.4. Role of ILK/GSK3β/C/EBPβ Axis During the Transcriptional Modification of CPT1A in an FA-Based Damage Model on Cultured HK2 Cells

As we observed from the previous results, CPT1A expression is implicated in the ILK-mediated regulation of mitochondrial loss of homeostasis. We further studied the transcriptional mechanisms implicated in the CPT1A transcriptional regulation in the HK2 model. As explained before, GSK3β is a downstream effector of ILK that has been related to transcriptional regulation of CPT1A expression [30]. Between the transcriptional factors that may change CPT1A expression, CCAAT/enhancer binding protein-β (C/EBPβ) is a candidate that has been already pointed to be a substrate for GSK3β [25,26]. To explore the ILK-mediated regulation of GSK3β and C/EBPβ on the FA-mediated downregulation of CPT1A, we used HK2 that were transfected with specific siRNAs for ILK (siILK), GSK3β (siGSK3β), or C/EBPβ (siC/EBPβ) prior to the FA or vehicle (CT) treatments. Scramble siRNAS were used as transfection controls when specific siRNAS were not used. The efficiency on GSK3β or C/EBPβ depletions using their specific siRNAs were verified by Western blot (Appendix A).

As in the mice model results shown in Figure 1E, Figure 9A shows that FA increases GSK3β phosphorylation on serine 9 in scramble-siRNA-transfected HK2 but not in siILK-transfected HK2. The total GSK3β contents were not different between groups. These data suggest that ILK mediates downstream phosphorylation of GSK3β in the HK2 model as in the in vivo model. In Figure 9B, it can be observed that FA reduces the expression of CPT1A in HK2, as already shown in Figure 8C. However, siGSK3β transfected HK2 reduced their CPT1A content at the same level as FA-treated cells. This result demonstrates the transcriptional regulation of CPT1A is necessarily mediated by the presence and activity of GSK3β.

We, therefore, studied the transcriptional activity of C/EBPβ as the candidate substrate for GSK3β activity in mediating CPT1A expression [25,30]. Figure 9C shows that the levels of C/EBPβ phosphorylated at Thr 235/188 (P-C/EBPβ), which is the transcriptionally active isoform, were decreased in FA-treated HK2. When ILK was depleted, P-C/EBPβ levels were duplicated in both experimental groups, siILK and siILK FA. It is important to notice that siILK FA fully recovered the downregulation of C/EBPβ activity observed in FA control cells.

As in Figure 9B,D shows that FA reduces the expression of CPT1A in HK2. However, siC/EBPβ-transfected HK2 reduced CPT1A content at the same level as FA-treated cells. This result demonstrates the transcriptional regulation of CPT1A is necessarily mediated by the presence and activity of C/EBPβ. These results confirm that FA-upregulated ILK downregulates the activity of GSK3β by direct or indirect phosphorylation on serine 9, which therefore reduces the C/EBPβ transcriptional activity on CPT1A expression. The transgenic depletion of ILK is nephroprotective in terms of preventing the mitochondrial dysfunction observed, probably due to the upregulation of CPT1A expression.

## 3. Discussion

The role of ILK in the pathogenesis of AKI and CKD has been previously described [5,6,7,8,9,10,11,12,13,14], but the underlying mechanism remains incompletely understood. In this study, we demonstrate that ILK overexpression contributes to mitochondrial dysfunction and increased autophagy in an FA-induced model of kidney damage. The FA-induced renal damage model is well established for studying the progression from AKI to CKD, as FA administration produces crystal formation in the tubular lumen, leading to tubular obstruction, cytokine release, oxidative stress, necrotic and apoptotic processes, and other CKD-related processes such as inflammation and fibrosis [2,21,31]. Our findings confirm that FA increases ILK expression and activity in both in vitro and in vivo models and that ILK depletion mitigates FA-induced tubular damage, fibrosis, and renal dysfunction.

Recent studies have highlighted mitochondrial dysfunction and impaired autophagy as critical biochemical pathways in the progression from AKI to CKD and renal fibrosis [22,33,34]. However, the precise mechanisms underlying these processes remain unclear. In this study, we explore the relationship between ILK, mitochondrial function, and autophagy in the context of renal disease. Our results indicate that FA-induced renal damage leads to mitochondrial dysfunction, characterized by decreased membrane potential, reduced expression of OXPHOS enzymes and CPT1A, and impaired activity of cytochrome C oxidase and citrate synthase. These findings are consistent with previous studies demonstrating mitochondrial impairment in kidney disease models [19,24,33,34]. A reduction in citrate synthase activity is frequently associated with mitochondrial mass loss [36]. In the present work, we have not measured the mitochondrial mass; however, since cytochrome C oxidase and citrate synthase activities were measured in isolated mitochondria, the observed reduction in enzymatic activity could be independent of mitochondrial mass loss. Additional experiments are needed to clarify this point.

Notably, ILK depletion restored mitochondrial function, suggesting a direct link between ILK overexpression and mitochondrial dysfunction in renal pathology. The regulation of mitochondrial function by ILK is not yet fully understood. Previous studies have suggested that the actin cytoskeleton, including ILK, plays a regulatory role in mitochondrial dynamics [37] and that ILK is crucial for the transcription of mitochondria-related genes and the maintenance of energy homeostasis in liver cells [38]. Our study provides novel evidence implicating ILK in mitochondrial dysfunction during kidney injury, underscoring its potential role as a therapeutic target for preserving mitochondrial function in renal disease.

Additionally, FA administration increased renal autophagic flux, as evidenced by the LC3BI/LC3BII ratio and p62 expression [17]. These effects were partially reversed by ILK depletion, suggesting that ILK plays a role in regulating autophagy. When dysregulated, autophagy can contribute to renal fibrosis and disease progression. Alterations in autophagy have been associated with various experimental models of renal damage [39]. In this context, maintaining a balance in renal autophagy is crucial for protecting against tubular injury [12,33,34], as excessive autophagy has been linked to the accumulation of fibrotic extracellular matrix (ECM) in the kidney [17,18]. Consistent with our findings, previous studies have described that ILK can inhibit autophagy through the ILK-downstream mediator AKT pathway while also promoting autophagy under stress conditions via AMP-activated protein kinase (AMPK) activation in various kidney diseases [40]. Furthermore, ILK-mediated inhibition of GSK3β activity may contribute to the upregulation of autophagic flux since GSK3β has been reported to activate mammalian target to rapamycin complex (mTORC), a key negative regulator of autophagy [41]. Further studies will be necessary to elucidate the underlying mechanisms by which ILK depletion affects autophagy in the context of renal disease.

Both mitochondrial dysfunction and increased autophagy can exacerbate renal damage in response to FA. The recent literature support the association between mitochondrial dysfunction, impaired autophagy, and renal fibrosis. However, the interplay between these processes remains unclear. Mitophagy, a selective form of autophagy that removes damaged mitochondria, is crucial for maintaining cellular homeostasis and preventing the accumulation of dysfunctional organelles that contribute to oxidative stress and inflammation. Given that ILK is involved in cytoskeletal organization and cellular signaling, it may interact with key regulators of mitophagy, such as AMPK, which is known to activate mitophagy under stress conditions [42]. While our study suggests that ILK overexpression influences both mitochondrial dysfunction and autophagy, future research is needed to determine whether ILK directly regulates mitophagy and its implications for renal disease progression. Among the mitochondrial markers regulated by ILK, CPT1A plays a crucial role in FAO, a major energy-producing pathway in renal tubules [20]. Our study reveals that CPT1A expression is significantly enhanced following ILK depletion in both in vivo and in vitro models. The proper functioning of CPT1A is essential for mitochondrial homeostasis, as it facilitates FAO and ATP production. Previous studies have shown that CPT1A expression is reduced in AKI and CKD models [22,23], while its overexpression protects against CKD-associated fibrosis, improving the mitochondrial membrane potential and preventing mitochondrial dysfunction [24]. ILK depletion not only increased CPT1A expression under basal conditions but also fully restored its levels in FA-treated cells, further reinforcing its protective role in mitochondrial function.

Therefore, we have analyzed the mechanism involved in regulating the expression of CPT1A by ILK. The presence of ILK can influence downstream effectors such as GSK3β by reducing its basal activity through phosphorylation at serine 9 [3]. After FA treatment, increased ILK levels led to the phosphorylation and subsequent reduction of GSK3β activity, which has been correlated with greater renal mitochondrial damage. It has been suggested that enhancing GSK3β activity may serve as a nephroprotective strategy to prevent renal fibrosis and mitochondrial dysfunction [15,16]. In fact, in vitro depletion of GSK3β using specific siRNAs reduced CPT1A expression to the same extent as FA treatment, confirming the role of GSK3β activity in the transcriptional regulation of CPT1A expression [30].

Importantly, our findings indicate that FA inhibits GSK3β activity in an ILK-dependent manner. Consequently, we investigated the possibility of a transcriptional pathway driven by ILK that modulates the mitochondrial marker CPT1A in renal tubules. Several transcription factors may regulate CPT1A expression. One well-established transcriptional substrate of GSK3β is the CCAAT/enhancer binding protein-β (C/EBPβ) [25,26], which is also a recognized transcription factor for CPT1A expression [25,30]. Moreover, C/EBPβ has been implicated in kidney function in several models of renal injury [27,28,29] and has been associated with mitochondrial biogenesis regulation during renal fibrosis in CKD models [21,43,44,45].

Our study demonstrates that FA treatment reduces the transcriptional activity of C/EBPβ. We confirmed that C/EBPβ depletion using specific siRNAs resulted in a significant downregulation of CPT1A, to the same extent as FA treatment. Furthermore, the role of ILK in regulating the C/EBPβ/CPT1A axis was confirmed using ILK-depleted cells, which exhibited increased transcriptional activity of C/EBPβ. Notably, FA treatment in ILK-depleted cells did not suppress this enhanced C/EBPβ activity.

In summary, our study describes a novel role for ILK in regulating renal mitochondrial function through the GSK3β/C/EBPβ/CPT1A axis in an FA-induced model of renal damage. To our knowledge, this is the first study identifying the ILK-GSK3β-C/EBPβ axis as a key regulator of the mitochondrial protective marker CPT1A. Therefore, we propose ILK as a potential therapeutic target to preserve renal mitochondrial homeostasis in kidney disease.

## 4. Materials and Methods

### 4.1. Animal Model of FA-Based AKI-to-CKD Renal Damage

The animal study protocol was approved by the Institutional Review Board (or Ethics Committee) of the University of Alcalá and Comunidad de Madrid (protocol code PROEX 160.7/22). Adult conditional ILK-deficient mice (cKD-ILK) were generated and used in our previous works [7]. Briefly, C57Bl/6 mice homozygous for floxed ILK flanked by loxP (LOX) were crossed with BALB/c strain mice carrying a CMV-driven hydroxytamoxifen-inducible (TX, Sigma-Aldrich, St. Louis, MO, USA) CreER (T) recombinase gene (CRE) globally expressed in all the tissues. Mice CRE-LOX (8-week-old) were injected intraperitoneally with 1.5 mg of TX or vehicle (VH, corn oil/ethanol, 9:1, Sigma-Aldrich) once per day for five consecutive days. Three weeks after the injections, tail DNA was genotyped by PCR with primers corresponding to the excised ILK gene (CCAGGTGGCAGAGGTAAGTA) or to non-excised ILK (CAAGGAATAAGGTGAGCTTCAGAA). The TX-treated CRE-LOX mice displaying successful depletion of ILK were termed ILK conditional-knockdown (cKD-ILK), and the VH-treated CRE-LOX without ILK depletion were termed Wild Type (WT). The cKD-ILK and wildtype (WT) littermates were injected intraperitoneally with a single dose of 250 mg/kg folic acid (FA) or vehicle as control (CT) [24,31,32]. After 15 days, animals were euthanized with pentobarbital overdose, and blood and renal samples were extracted, preserved, and processed accordingly.

### 4.2. Serum Creatinine and Blood Urea Nitrogen Determination

Serum creatinine and blood urea nitrogen (BUN) were determined with commercial kits according to the manufacturer’s protocol (Cayman Chemicals, Ann Arbor, MI, USA and Invitrogen, TermoFisher Scientifc, Waltham, MA, USA), respectively, measured in mg/dl.

### 4.3. Renal Histopathological Analysis

Kidney tissue was fixed in 4% Paraformaldehyde, dehydrated, and embedded in paraffin. Kidney sections (3 μm) were stained with Hematoxylin-eosin according to the manufacturer’s instructions (Casa Álvarez Cientific Material S.A., Madrid, Spain). Between 4 and 10 random fields from each renal cortex section were photographed. Morphological analyses were performed blindly by two different experienced pathologists, according to our previous reports [5,6]. Briefly, the tubular damage parameters scored and counted were dilatation, atrophy, and pus casts. Damaged tubules were counted and expressed as a percentage of the total number of tubules per area. Interstitial collagen content as a feature of fibrosis was assessed by Sirius red staining (Polysciences, Warrington, PA, USA) in more paraffin-embedded renal cortex sections, as described previously [5,6]. Briefly, paraffin-embedded kidney samples were incubated with Sirius red solution for 60 min, washed in acidified water, dehydrated in ethanol, cleared in xylene, and mounted in DPX. Between 4 and 10 random fields from each renal cortex section were photographed, and red intensity per photographed areas were determined using ImageJ plus software 1.54g, Java 1.8.0_345, (National Institutes of Health, Bethesda, MD, USA).

### 4.4. Cell Culture and In Vitro FA Treatment

The human kidney tubular epithelial cell line (HK2; CRL-2190) was purchased from the American Type Culture Collection (ATCC, distributed by LGC Standards, Barcelona, Spain). Cells were routinely cultured in 95% O_2_ and 5% CO_2_ at 37 °C. For the experiments, cells were grown on 6-well plates with DMEM-F12 (Sigma-Aldrich, St. Louis, MO, USA), 10% fetal bovine serum (FBS; Sigma-Aldrich, St. Louis, MO, USA), 100 U/mL Penicillin-streptomycin (P/S; TermoFisher Scientific, Waltham, MA, USA) and supplemented with 5 ng/mL Endothelial Growth Factor (EGF; TermoFisher Scientific, Waltham, MA, USA). Two days after cells reached confluence, cells were transfected with 10 nM of ILK, GSK3β or C/EBPβ specific siRNA or Silencer-negative control (Scrambled RNA) using Metafectene (Biontex, Munich, Germany). After 24 h incubation with the RNA complex, 1 mL of medium containing FBS was added overnight, and then cells were deprived with DMEM-F12 without FBS during 24 h. After this, cells were treated with 10 mM FA (Sigma-Aldrich, St. Louis, MO, USA) for 24 h. Finally, we removed the FA from the plates, and cells were washed with phosphate buffer saline (PBS; Thermo-Fisher, Waltham, MA, USA).

### 4.5. Renal Mitochondrial Extracts and Determinations of Cytochrome C Oxidase and Citrate Synthase Activities

Mitochondria and cytosolic extracts were isolated from renal tissues or cultured HK2 cells using a commercial Mitochondria/Cytosol Fractionation Kit (ab65320, Abcam, Cambridge, UK) and used to determine the activities of Cyt C oxidase and Citrate synthase using commercial Kits. Briefly, cells were removed after treatment with FA, incubated with ice-cold 1× Cytosol extraction buffer mix for 10 min, and were collected and centrifugated at 700× *g* for 10 min at 4 °C. The supernatant was centrifuged at 10,000× *g* for 30 min at +4 °C and collected cytosolic fraction and mitochondrial fraction. After isolation mitochondria of renal tissue or HK-2 cells, cytochrome C oxidase activity was determined using a Cytochrome C Oxidase Assay Kit (ab239711, Abcam, Cambridge, UK). The assay was performed with 50 µg protein, and the cytochrome C oxidase activity was displayed as μmol Cytochrome C reduced/min. Citrate synthase activity was determined using a Citrate Synthase Activity Assay Kit (ab239712, Abcam, Cambridge, UK). The assay was performed with 30 µg protein, and the citrate synthase activity was displayed as nmol/min/μL.

### 4.6. Mitochondrial Membrane Potential Assay

Mitochondrial membrane potential was determined in living HK2 cells by Tetramethylrhodamine methyl ester (TMRM), a dye that penetrates cells and accumulates within active mitochondria with intact membrane potentials (Invitrogen, ThermoFisher Scientifc, Waltham, MA, USA). Cells were stained afterwards with 4′,6-diamidino-2-phenylindole (DAPI), and visualized under a confocal microscope (Leica SP5 confocal microscope (Leica Microsystems, Wetzlar, Germany)). TMRM fluorescence was detected in the red channel, while DAPI fluorescence was visualized in the blue channel. Pictures show TMRM in red and DAPI in blue. Mitochondrial membrane potential was quantified by measuring the ratio of red fluorescence intensity, normalized to the number of cells, as determined by DAPI staining [35]. For quantitative analysis, images were processed using ImageJ software 1.54g, Java 1.8.0_345 (NIH, Bethesda, MD, USA). Fluorescence intensity thresholds were standardized across all images to maintain consistency in quantification. A decrease in TMRM fluorescence indicated mitochondrial depolarization, suggesting impaired mitochondrial function. In contrast, high TMRM retention reflected preserved membrane potential and functional mitochondria.

### 4.7. Western Blots

Cells or tissues were homogenized in lysis buffer (10 mM Tris–HCl, pH 7.6; 1% Triton X-100; 1 mM EDTA; 0.1% sodium deoxycholate) supplemented with protease and phosphatase inhibitors (Complete and PhosSTOP from Roche, Basel, Switzerland). Protein concentrations were determined by DC-protein assay (Bio-Rad, Hercules, CA, USA). Equal protein amounts were separated on SDS-polyacrylamide gels and transferred to 0.2 μm PVDF membranes (Bio-Rad, Hercules, CA, USA). Membranes were blocked to 3% Bovine Serum Albumin (BSA; Sigma-Aldrich, USA) and were incubated with primary and secondary antibodies afterwards. Primary antibodies used were against ILK, GSK3β, P-GSK3β (Ser9), SQSTM1/p62 and LC3BI/II (Cell Signaling Technology, Danvers, MA, USA); C/EBPβ and P-C/EBPβ (Thr235/188) (Antibodies.com, Stockholm, Sweden); FN, COL1A1 and CPT1A (Abcam, Cambridge, UK), and finally GAPDH (Sigma-Aldrich, St. Louis, MO, USA). Antibodies cocktail kit (Abcam, Cambridge, UK) was used to detect the following Oxidative phosphorylation (OXPHOS)-related enzymes content: ATP Synthase (ATP5A1), Coenzyme Q (UQRC2), mitochondrial Cytochrome (Cyt) C oxidase subunit 1 (MT-CO1), Succinate Dehydrogenase (SDHB) and NADPH Dehydrogenase (NDUFB8) [19,33,34,35]. Secondary antibodies used were against rabbit or mice (Merck-Millipore, Burlingto, MA, USA; Dako, Glostrup, DNK), respectively. Immunoblots were detected by chemiluminescence (Pierce ECL Western Blotting Substrate; ThermoFisher Scientific, MA, USA) and imaged with ImageQuant LAS 500 System (General Electric Healthcare, Chicago, IL, USA). Densitometry analyses were performed using ImageJ software (National Institutes of Health, Bethesda, MD, USA).

### 4.8. Reverse Transcription–Quantitative Polymerase Chain Reaction (RT-qPCR)

Total RNA from cell samples was extracted with Nyzol (Nzytech, Lisboa, Portugal). The rest of the reagents, products, and equipment were used from Thermo Fisher Scientific, MA, USA. Equal amounts of RNA were transcribed to cDNA with High-Capacity cDNA RT Kit, and RT-qPCR was performed as described previously [6]. TaqMan gene expression assays were used to quantify ILK (Hs00177914_m1), FN (Hs01549976_m1), and COL1A1 (Hs00164004_m1) and were normalized to β-actin (Hs01060665_g1).

### 4.9. Statistical Analysis

All statistical analyses were performed using the GraphPad Prism version 5.00 (San Diego, CA, USA). The results are presented as the mean  ±  standard error of the mean (SEM). All experiments were repeated at least four times (the number of experiments is provided in the legends on the figures). The normality of the value distributions was assessed by the Kolmogorov–Smirnov test. In those cases, a two-way ANOVA test was performed, followed by a post-hoc analysis. Otherwise, Kruskal–Wallis tests, followed by Mann–Whitney or Wilcoxon post-tests, were used with the Bonferroni correction. Differences in mean values were considered statistically significant at a probability level of less than 5% (*p* < 0.05).

## 5. Conclusions

FA increases the expression of ILK, which plays a significant role in various cellular processes, including mitochondrial function and energy metabolism. During FA-induced renal injury, increased ILK expression leads to enhanced phosphorylation and subsequent inhibition of GSK3β. Inhibited GSK3β, which has been directly implicated during renal damage, suppresses the transcriptional activity of C/EBPβ, a crucial factor in CPT1A expression. CPT1A, along with OXPHOS enzymes and autophagy-related markers, are downregulated during FA-induced tubular fibrosis. Here, we demonstrate that these alterations are influenced by ILK presence. Consequently, we propose ILK as a novel pharmacological target for preserving renal mitochondrial homeostasis. Targeting ILK could potentially improve mitochondrial health and function, offering a therapeutic strategy to mitigate the symptoms or progression of diseases associated with mitochondrial dysfunction.

## Figures and Tables

**Figure 1 ijms-26-01861-f001:**
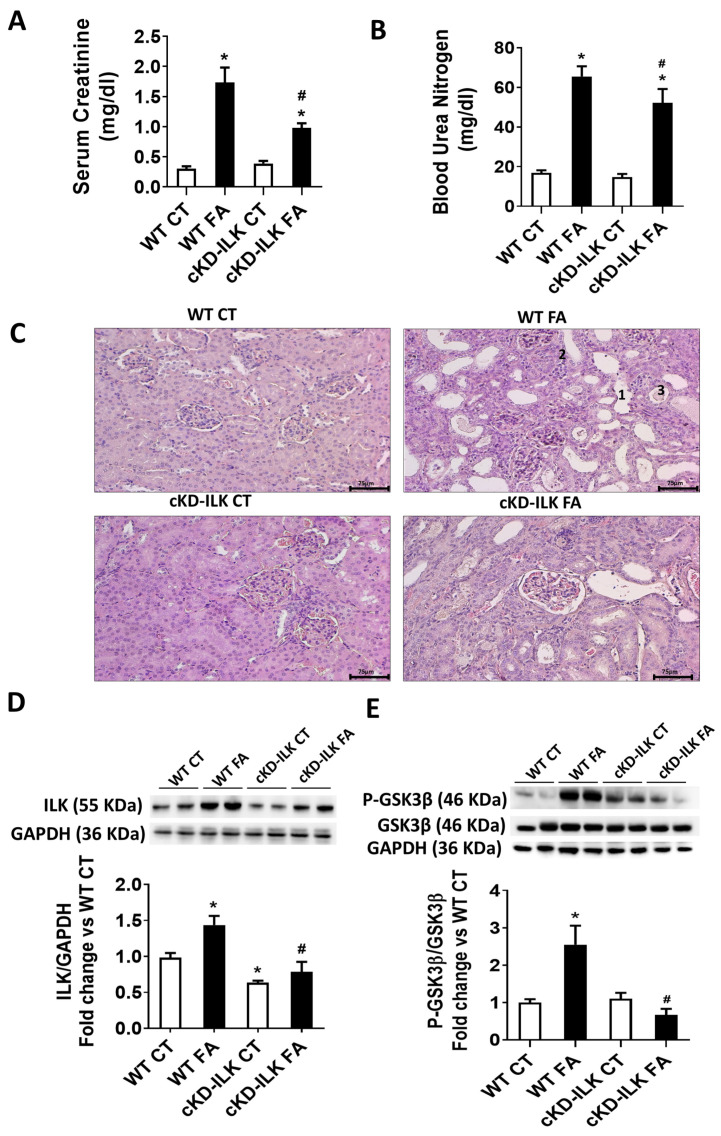
FA-induced renal damage on mice is prevented by the lack of ILK. Wild type (WT) and ILK conditional-knockdown (cKD-ILK) mice were injected intraperitoneally with a single dose of folic acid (FA, 250 mg/Kg) or vehicle (CT). After 15 days of treatment, kidneys and blood were extracted. (**A**) Serum creatinine and (**B**) blood urea nitrogen (BUN) were determined by colorimetric assay kits. (**C**) Renal slides were stained with Hematoxylin-eosin and histologically analyzed. Representative renal cortex images are shown. Scale bars: 75 μm. Numbers indicate FA-mediated tubular dilation (1), atrophy (2), and pus casts (3). The analysis of renal damage for each experimental group is compiled in Table 1. (**D**) ILK total protein content and (**E**) Total and phosphorylated GSK3β isoform at Ser-9 (P-GSK3β) protein content were determined by Western blot. Representative blots are shown. Protein densitometries were normalized to endogenous GAPDH or GSK3β total contents, respectively. The results are expressed as fold change of WT CT and are the mean ± SEM from 8 WT CT, 5 WT FA, 8 cKD-ILK CT, 4 cKD-ILK FA. * *p* < 0.05 vs. WT CT; # *p* < 0.05 vs. WT FA.

**Figure 2 ijms-26-01861-f002:**
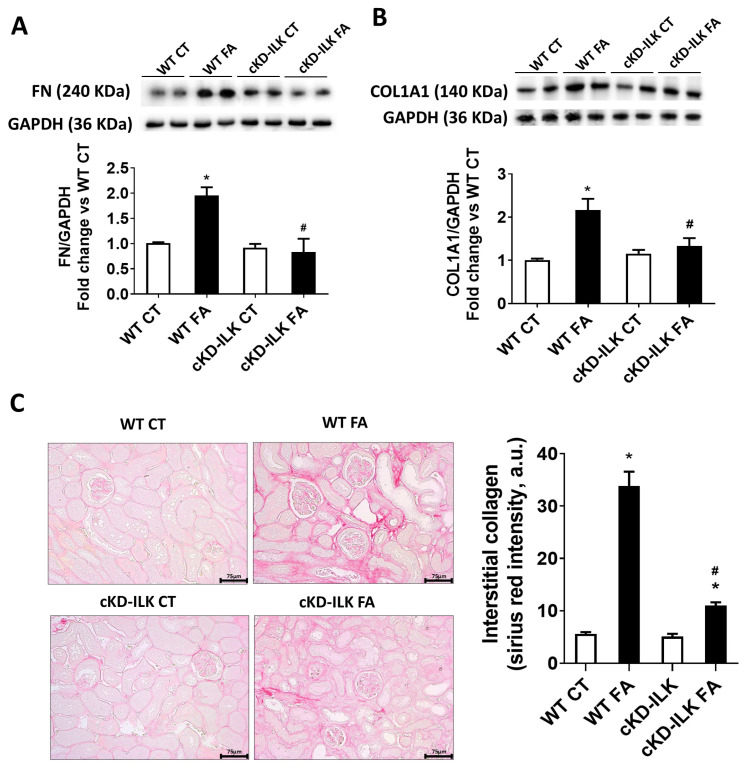
FA-induced renal fibrosis is prevented by the lack of ILK. Wild type (WT) and ILK conditional-knockdown (cKD-ILK) mice were injected intraperitoneally with a single dose of folic acid (FA, 250 mg/Kg) or vehicle (CT). After 15 days of treatment, kidneys were extracted. (**A**) Fibronectin (FN) and (**B**) Collagen type I (COL1A1) total protein contents were determined by Western blot. Representative blots are shown. Protein densitometries were normalized to endogenous GAPDH, and the relative fold changes vs. WT CT are represented. (**C**) Interstitial collagen content as a feature of fibrosis was assessed by Sirius red staining in fixed, dehydrated, and paraffin-embedded renal cortex sections. Representative microphotographs are shown, Scale bars: 75 μm. The red intensity in each photographed area was quantified and represented in arbitrary units (a.u.). Values are the mean ± SEM from 8 WT CT, 5 WT FA, 8 cKD-ILK CT, 4 cKD-ILK FA. * *p* < 0.05 vs. WT CT; # *p* < 0.05 vs. WT FA.

**Figure 3 ijms-26-01861-f003:**
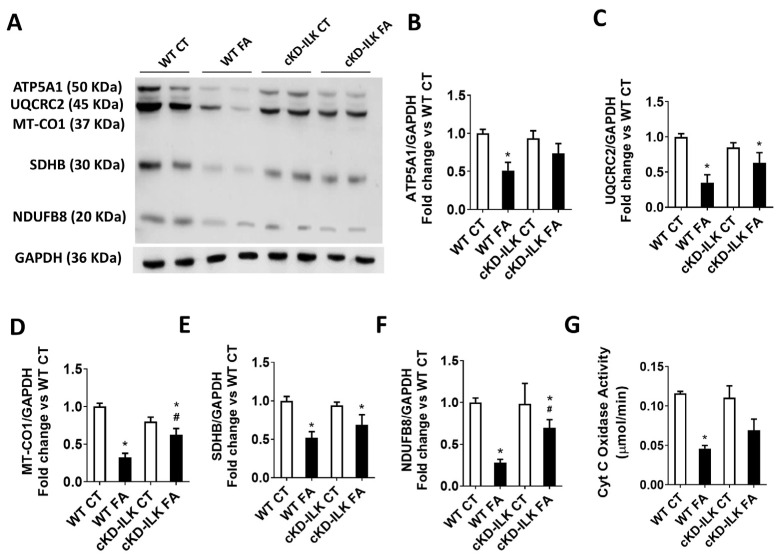
The lack of ILK prevents the FA-induced reduction of renal oxidative phosphorylation (OXPHOS)-related enzymes’ expression and activity. Wild type (WT) and ILK conditional-knockdown (cKD-ILK) mice were injected intraperitoneally with a single dose of folic acid (FA, 250 mg/Kg) or vehicle (CT). After 15 days of treatment, the renal protein extracts were obtained. (**A**) A representative immunoblots of OXPHOS-related enzymes. (**B**) ATP Synthase (ATP5A1), (**C**) Coenzyme Q (UQRC2), (**D**) mitochondrial Cytochrome (Cyt) C oxidase subunit 1 (MT-CO1), (**E**) Succinate Dehydrogenase (SDHB), and (**F**) NADPH Dehydrogenase (NDUFB8) total protein contents were determined by Western blot. Protein densitometries were normalized to endogenous GAPDH. The results are expressed as the fold change of WT CT, from 8 WT CT, 5 WT FA, 8 cKD-ILK CT, and 4 cKD-ILK FA. (**G**) Mitochondrial Cyt C Oxidase activity was measured using a colorimetric assay kit in renal mitochondrial extracts after 8 min of initiating the reaction following the commercial kit instructions. The data are μmol Cyt C reduced/min from three different experiments. Results are the mean ± SEM. * *p* < 0.05 vs. WT CT; # *p* < 0.05 vs. WT FA.

**Figure 4 ijms-26-01861-f004:**
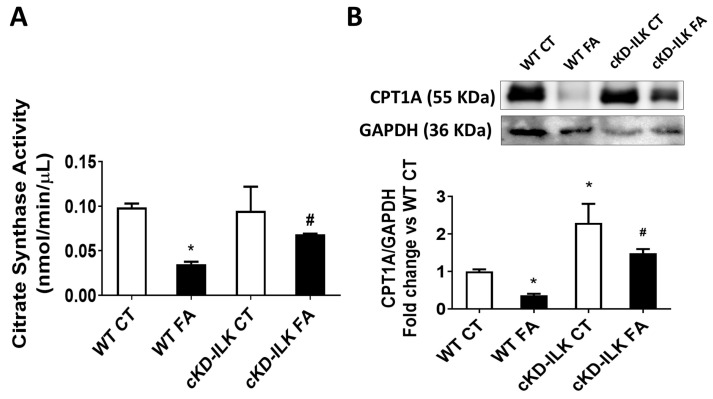
The lack of ILK prevents the FA-induced reduction of renal mitochondrial citrate synthase activity and CPT1A expression. Wild type (WT) and ILK conditional-knockdown (cKD-ILK) mice were injected intraperitoneally with a single dose of folic acid (FA, 250 mg/Kg) or vehicle (CT). After 15 days of treatment, the renal protein extracts were obtained. (**A**) Citrate synthase activity was measured using a colorimetric assay kit in renal mitochondrial extracts 5 min after initiating the reaction. The data are nmol/min/μL from three independent experiments. (**B**) CPT1A total protein content was determined by Western blot. Representative blots are shown. Protein densitometries were normalized to endogenous GAPDH. The results are expressed as fold change of WT CT and are the mean ± SEM from 8 WT CT, 5 WT FA, 8 cKD-ILK CT, 4 cKD-ILK FA. * *p* < 0.05 vs. WT CT; # *p* < 0.05 vs. WT FA.

**Figure 5 ijms-26-01861-f005:**
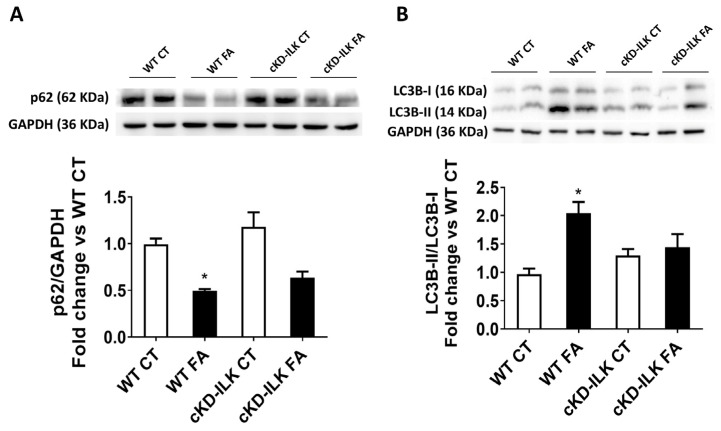
The lack of ILK prevents the FA-induced increase of renal autophagy markers. Wild type (WT) and ILK conditional-knockdown (cKD-ILK) mice were injected intraperitoneally with a single dose of folic acid (FA, 250 mg/Kg) or vehicle (CT). After 15 days of treatment, the renal protein extracts were obtained. (**A**) p62 and (**B**) Light chain 3-B (LC3B) isoforms I and II total protein contents were determined by Western blot. Representative blots are shown. Protein densitometries and rates for LC3B isoforms I and II were normalized to endogenous GAPDH. The results are expressed as fold change of WT CT and are the mean ± SEM from 8 WT CT, 5 WT FA, 8 cKD-ILK CT, 4 cKD-ILK FA. * *p* < 0.05 vs. WT CT.

**Figure 6 ijms-26-01861-f006:**
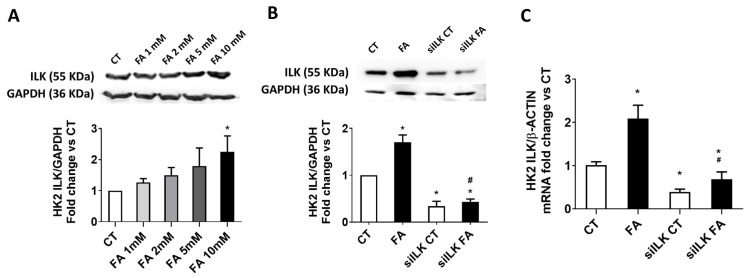
ILK overexpression in an FA-based damage model on cultured HK2 cells. (**A**) HK2 cells were treated with different FA doses for 24 h. (**B**,**C**) HK2 cells were transfected for 24 h with a specific siRNA against ILK (siILK) or scramble siRNA as transfection control and treated afterwards with 10 mM FA or vehicle (CT) for another 24 h. (**A**,**B**) ILK total protein content determined by Western blot. Representative blots are shown. Protein densitometries were normalized to endogenous GAPDH. The results are expressed as fold change of CT and are the mean ± SEM from four independent experiments (**C**) ILK mRNA levels determined by RT-qPCR and normalized to endogenous β-actin mRNA content. The results are expressed as fold change of CT and are the mean ± SEM from three independent experiments * *p* < 0.05 vs. CT; # *p* < 0.05 vs. FA.

**Figure 7 ijms-26-01861-f007:**
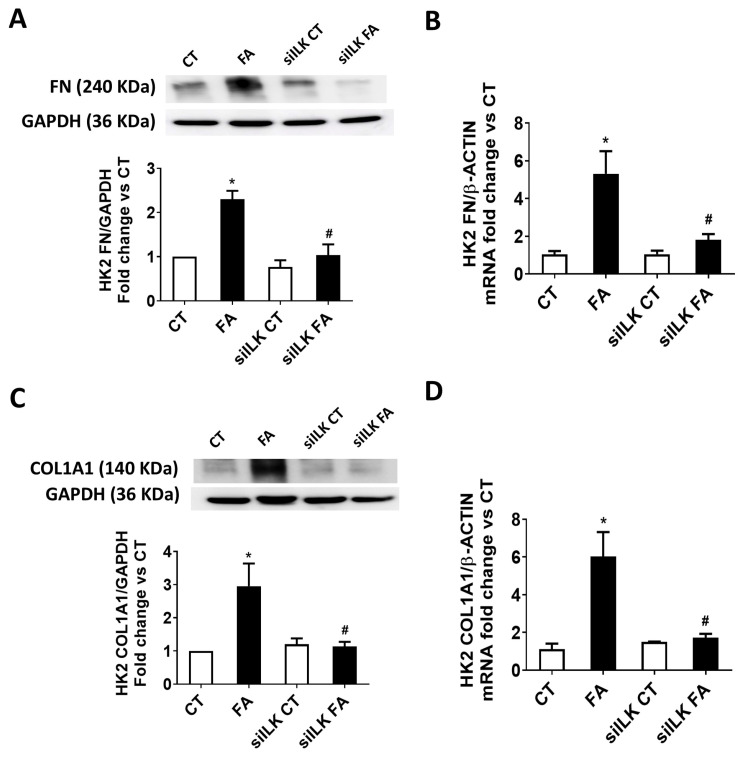
The lack of ILK prevents FA-induced fibrosis in cultured HK2 cells. HK2 cells were transfected for 24 h with a specific siRNA against ILK (siILK) or Scramble siRNA as transfection control and treated afterwards with 10 mM FA or vehicle (CT) for another 24 h. Fibronectin (FN) (**A**) total protein content or (**B**) mRNA and Collagen type I (COL1A1) (**C**) Total protein content or (**D**) mRNA were determined by Western blot or RT-qPCR, respectively. Representative blots are shown. Protein densitometries were normalized to endogenous GAPDH, and gene expression normalized to endogenous β-actin mRNA content. The results are expressed as fold change of CT and are the mean ± SEM from four independent experiments * *p* < 0.05 vs. CT; # *p* < 0.05 vs. FA.

**Figure 8 ijms-26-01861-f008:**
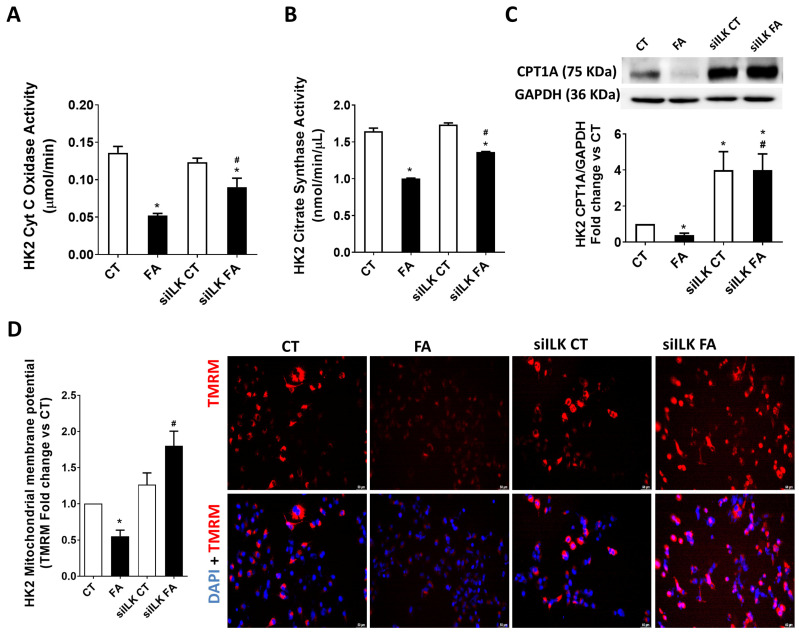
The lack of ILK prevents the FA-induced reduction of mitochondrial activity and CPT1A expression in cultured HK2 cells. HK2 cells were transfected for 24 h with a specific siRNA against ILK (siILK) or scramble siRNA as transfection control and treated afterwards with 10 mM FA or vehicle (CT) for another 24 h. (**A**) Mitochondrial Cyt C oxidase and (**B**) Citrate synthase activities from three independent experiments were measured in mitochondrial extracts from transfected HK2 cells after FA by colorimetric assay kits. Results shown are μmol Cyt C reduced/min after 8 min and citrate nmol/min/μL after 7 min of initiating the enzymatic reactions, respectively. (**C**) CPT1A total protein content was determined by Western blot. Protein densitometries were normalized to endogenous GAPDH. (**D**) Representative confocal pictures and analysis of mitochondrial membrane potential analyzed as intracellular presence of Tetramethylrhodamine methyl ester (TMRM, red fluorescence) and normalized to total cell count (nuclei count stained with DAPI in blue). The results are expressed as fold change of CT and are the mean ± SEM from five independent experiments. * *p* < 0.05 vs. CT; # *p* < 0.05 vs. FA.

**Figure 9 ijms-26-01861-f009:**
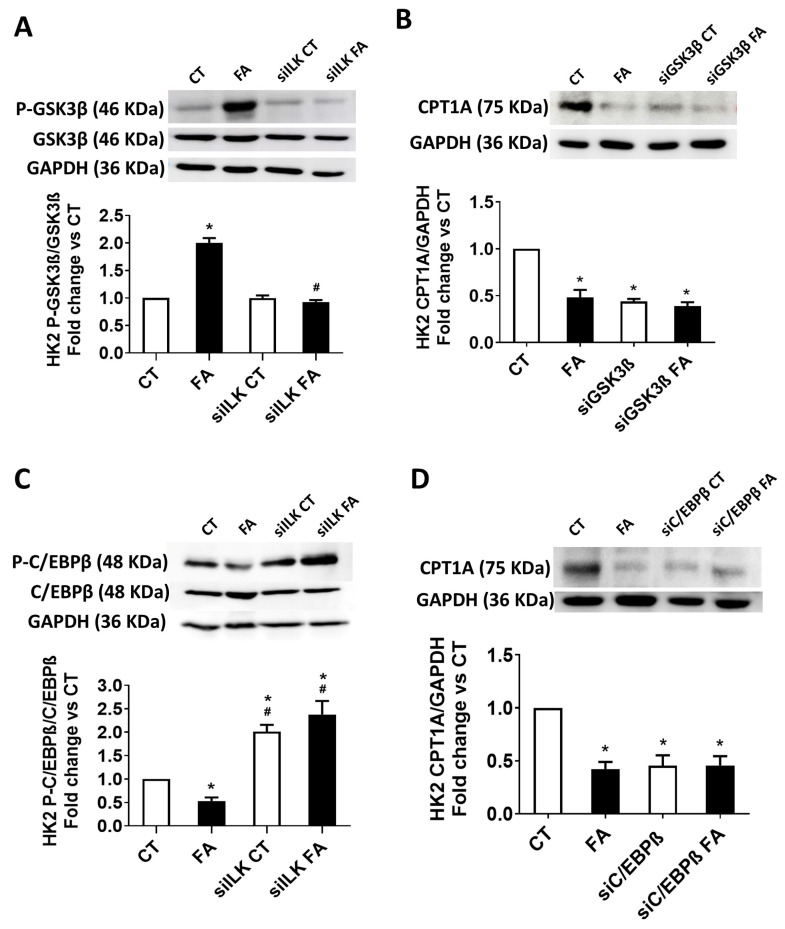
Role of ILK/GSK3β/C/EBPβ axis on mitochondrial CPT1A expression in an FA-based damage model on cultured HK2 cells. HK2 cells were transfected for 24 h with a specific siRNA against ILK (siILK), GSK3β (siGSK3β), C/EBPβ (siC/EBPβ), or scramble siRNA as transfection control and treated afterwards with 10 mM FA or vehicle (CT) for another 24 h. (**A**) Total and phosphorylated GSK3β isoform at Ser-9 (P-GSK3β) protein contents from siILK-transfected cells, (**B**) total CPT1A protein content from siGSK3β-transfected cells from four independent experiments, (**C**) total and phosphorylated C/EBPβ phosphorylation isoform at Thr235/188 (P-C/EBPβ) protein contents from siILK-transfected cells from six independent experiments and (**D**) total CPT1A protein content from siC/EBPβ-transfected cells from five independent experiments, were all determined by Western blot. Protein densitometries were normalized to total GSK3β, GAPDH, C/EBPβ, or GAPDH, respectively. The results are expressed as fold change of CT and are the mean ± SEM. * *p* < 0.05 vs. CT; # *p* < 0.05 vs. FA.

**Table 1 ijms-26-01861-t001:** Histological tubular damage analysis from mice subjected to the FA model.

Experimental Group	Tubular Dilatation (%)	Tubular Atrophy (%)	Pus Casts in Tubules (%)
WT CT	3.2 ± 0.3	1.2 ± 0.9	0 ± 0
WT FA	26.2 ± 3.2 *	33.4 ± 0.9 *	18.3 ± 2.9 *
cKD-ILK CT	4.5 ± 0.7	1 ± 0.6	0 ± 0
cKD-ILK FA	10.9 ± 1.3 *#	21.8 ± 1 *#	11.4 ± 2.15 *#

Wild type (WT) and ILK conditional-knockdown (cKD-ILK) mice that were injected intraperitoneally with a single dose of folic acid (FA, 250 mg/Kg) or vehicle (CT), as described in Figure 1, were sacrificed, and kidneys were fixed, dehydrated, and paraffin-embedded. Sections of the kidneys were stained with hematoxylin-eosin. Several random fields from each renal cortex section were photographed, and morphological analyses were performed blindly by two different experienced pathologists. The tubular damage parameters scored and counted were dilation, atrophy, and pus casts. Damaged tubules were counted and expressed as a percentage of the total number of tubules per area. Values are the mean ± SEM from 8 WT CT, 5 WT FA, 8 cKD-ILK CT, 4 cKD-ILK FA. * *p* < 0.05 vs. WT CT; # *p* < 0.05 vs. WT FA.

## Data Availability

Data is contained within the article. The original contributions presented in the study are included in the article; further inquiries can be directed to the corresponding author.

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
