# Peer review of "Integrin-Linked Kinase (ILK) Promotes Mitochondrial Dysfunction by Decreasing CPT1A Expression in a Folic Acid-Based Model of Kidney Disease"

_ijms, 2025, doi:10.3390/ijms26051861_

Round 1

Reviewer 1 Report

Comments and Suggestions for Authors

In the article entitled "Integrin-bound kinase (ILK) promotes mitochondrial dysfunction by decreasing expression of CPT1A in a folic acid 3 model of kidney disease", the authors showed the effect of ILK on a folic acid-induced damage model. The manuscript is relevant; however, it needs to be improved. In general, the entire results and discussion section needs to be rewritten. Too many flaws in both sections do not allow the necessary fluency to understand the results.

1. Please include the molecular weights of the wb proteins shown in the representative images of the main manuscript.
2. The description of the results is not clear. On several occasions the authors did not describe precisely what was found or what differences existed between the groups studied. For example, the description of OXPHOS is very technical, and therefore, all results are presented in a technical manner. Same for activities
3. Following this last comment, there is no sequence between the description of the data obtained,d and this is a problem because it does not allow to make a connection with what is being observed and simulate isolated data.
4. Why did the authors decide to evaluate citrate synthase activity? This should be clear at the beginning of the results section. Is the reduction in citrate synthase activity related to mitochondrial mass loss? Please discuss this in the discussion section.
5. What is the relationship between citrate synthase and CPT1A? Please explain why these data are presented together
6. Authors should conclude each section result. This allows the reader to get closer to the study’s understanding.
7.  The discussion is vague and does not precisely "discuss" the relevant results of the manuscript. There is no comparison with other studies and although it is understood that this is the first time that ILK relevancy has been demonstrated in the model of AF and mitochondrial dysfunction, the importance of the study needs to be justified.

8.  The authors need to delve into the relationship of ILK/GSK3b and mitochondrial dysfunction. Is ILK related to mitochondrial mass loss? What about autophagy and GSK3b.

9. If the authors claim that autophagy is relevant in the research, what about mitophagy? Mitophagy is even more relevant for mitochondria degradation, and previous studies have pointed to the alteration of PINK and Parkin in the FA model.

Minor comments:

1. Please rewrite the following sentence: " To our knowledge, this is the first time that ILK has been implicated in regulating such a relevant marker of mitochondrial homeostasis." it is difficult to understand.

2.  Define all abbreviations the first time they are used. A list of abbreviations is required.

3. Please describe how mitochondria were obtained.

4. Mitochondria from culture cells is commonly difficult to obtain; how can authors obtain mitochondria from HK-2 cells?

Comments on the Quality of English Language

The English need to be improved 

Author Response

COMMENTS 1

In the article entitled "Integrin-bound kinase (ILK) promotes mitochondrial dysfunction by decreasing expression of CPT1A in a folic acid 3 model of kidney disease", the authors showed the effect of ILK on a folic acid-induced damage model. The manuscript is relevant; however, it needs to be improved. In general, the entire results and discussion section needs to be rewritten. Too many flaws in both sections do not allow the necessary fluency to understand the results.

RESPONSE 1

We sincerely appreciate the reviewers´ valuable suggestions for improving our article. Following their recommendations, we have rearranged several sections of the original text. Please, find the changes in red font throughout the entire manuscript. We double checked the English grammar with a bilingual co-worker and changed the text accordingly.

COMMENTS 2

Please include the molecular weights of the wb proteins shown in the representative images of the main manuscript.

RESPONSE 2

As suggested by the reviewer, we changed the figures and included the molecular weight of each protein shown

COMMENTS 3

The description of the results is not clear. On several occasions the authors did not describe precisely what was found or what differences existed between the groups studied. For example, the description of OXPHOS is very technical, and therefore, all results are presented in a technical manner. Same for activities

RESPONSE 3 We thank the reviewer for bringing up this matter. To bring overall clarity and completeness, we tried to improve the descriptions in results and discussion sections in the revised manuscript. We carefully changed the entire results and discussion sections texts, to describe better and precisely what is shown in each figure and why we studied it.

COMMENTS 4

Following this last comment, there is no sequence between the description of the data obtained,d and this is a problem because it does not allow to make a connection with what is being observed and simulate isolated data.

RESPONSE 4

As commented on the previous question, we followed the reviewer’s suggestion to better express the connection of results. In each results description, we tried to explain also the importance of the findings.

COMMENTS 5

Why did the authors decide to evaluate citrate synthase activity? This should be clear at the beginning of the results section. Is the reduction in citrate synthase activity related to mitochondrial mass loss? Please discuss this in the discussion section.

RESPONSE 5

We sincerely thank the reviewer for raising these points.  The citrate synthase is a limiting enzyme of tricarboxylic acid cycle, frequently used to assess mitochondrial function and mitochondrial mass. In response to the request, more information is included in results and discussion sections. (lines 200-212 and line 395)

COMMENTS 6

What is the relationship between citrate synthase and CPT1A? Please explain why these data are presented together

RESPONSE 6

Renal tubular cells primarily rely on fatty acid oxidation (FAO) for ATP production. This process occurs in the mitochondria after long-chain fatty acids enter, mediated by CPT1A. Subsequently, acetyl-CoA is recycled through the tricarboxylic acid (TCA) cycle in successive rounds. Citrate synthase acts as a limiting enzyme in the TCA cycle. Therefore, we decided to present both sets of data together.

COMMENTS 7

Authors should conclude each section result. This allows the reader to get closer to the study’s understanding.

RESPONSE 7

Following the reviewer's suggestion, we have included a statement at the end of each results section to summarize the main findings.

COMMENTS 8

The discussion is vague and does not precisely "discuss" the relevant results of the manuscript. There is no comparison with other studies and although it is understood that this is the first time that ILK relevancy has been demonstrated in the model of AF and mitochondrial dysfunction, the importance of the study needs to be justified.

RESPONSE 8

We agree with the reviewer's suggestion. Consequently, we have rewritten the entire discussion section, emphasizing the significance of the results obtained regarding the role of overexpressed ILK in mitochondrial dysfunction in renal disease.

COMMENTS 9

 The authors need to delve into the relationship of ILK/GSK3b and mitochondrial dysfunction. Is ILK related to mitochondrial mass loss? What about autophagy and GSK3b.

RESPONSE 9

The reviewer is correct. We have demonstrated the role of the ILK/GSK3β axis in mitochondrial dysfunction, as depletion of ILK improves mitochondrial function. Specifically, the ILK/GSK3β axis regulates the transcription of the CPT1A gene through the C/EBPβ transcription factor. CPT1A is a crucial enzyme in mitochondrial function, as its activity limits fatty acid oxidation.

We have now clarified this important point in the discussion section and added new references. Unfortunately, we have not measured mitochondrial mass under our experimental conditions, so we cannot conclude if there is a relationship between them. We have not measured the direct role of GSK3β in autophagy. However, we discuss the potential role of GSK3β in mediating the effect of ILK on autophagy, as suggested by the literature.

Please refer to lines 403 to 436 (citations 37 to 42).

COMMENTS 10

If the authors claim that autophagy is relevant in the research, what about mitophagy? Mitophagy is even more relevant for mitochondria degradation, and previous studies have pointed to the alteration of PINK and Parkin in the FA model.

RESPONSE 10

We thank the reviewer for the comment. Indeed, the regulation of mitophagy by ILK would be an interesting issue to address in relation to renal damage. Unfortunately, we have not measured it, but we have commented on the possible relevance of mitophagy in our model of FA-induced renal damage in the discussion section. Please, refer to lines 430 to 439.

COMMENTS 11

Minor comments: 1. Please rewrite the following sentence: " To our knowledge, this is the first time that ILK has been implicated in regulating such a relevant marker of mitochondrial homeostasis." it is difficult to understand.

RESPONSE 11

As the Reviewer observed, the sentence had no sense.  We deleted that sentence, since it was difficult to understand, and it did not provide any relevant information to the paragraph.

COMMENTS 12

Minor comments: 2.  Define all abbreviations the first time they are used. A list of abbreviations is required.

RESPONSE 12

We define all the abbreviations the first time they are used in the revised manuscript. We are sorry if in the original manuscript we missed defining some of them. We are not listing the abbreviations because the editorial guidelines for authors do not allow to add a list of abbreviations.

COMMENTS 13

Minor comments: 3. Please describe how mitochondria were obtained.

RESPONSE 13

We appreciate that the reviewer asks for a better explanation of the methodology used to obtain mitochondrial extracts.

We used a commercial kit from Abcam (Mitochondria/Cytosol Fractionation Kit, catalog number ab65320) to obtain mitochondrial and cytosol extracts from HK2 cells as well as from renal extracts. We followed the manufacturer’s protocol. We detailed the procedure in a differentiated sub-section inside the methodological descriptions (now the section is 4.5. Renal mitochondrial extracts and determinations of Cythochrome C oxidase and Citrate synthase activities)

COMMENTS 14

Minor comments: 4. Mitochondria from culture cells is commonly difficult to obtain; how can authors obtain mitochondria from HK-2 cells?

RESPONSE 14

As mentioned in the previous question, we detailed the procedure with more clarity in a new subsection in the methods section (4.5. Renal mitochondrial extracts and determinations of Cythochrome C oxidase and Citrate synthase activities)

We used a commercial kit from Abcam (catalog number ab65320) to obtain mitochondrial and cytosol extracts from HK2 cells as well as from renal extracts. We followed the manufacturer’s protocol.  Initially, cells were harvested and washed with cold phosphate-buffered saline (PBS) to remove any residual culture medium. The cell pellet was then resuspended in the provided cell lysis buffer (Cytosol extraction buffer), containing protease inhibitors, and incubated on ice for 10 minutes to facilitate the disruption of the cell membrane, and then centrifuged at 700 x g for 30 min at +4°C. The supernatant, which contains the cytosolic fraction, was carefully collected. For mitochondrial enrichment, the supernatant was further centrifuged at a higher speed (10,000 x g for 30 min at +4°C) to isolate the mitochondrial fraction and used afterwards in the determination of enzymatic activities, based in commercial kits, and using the protocols provided by the manufacturers (both assays kits were from Abcam)

COMMENTS 15

Comments on the Quality of English Language.

The English need to be improved

RESPONSE 15

We double checked the English grammar with a bilingual co-worker and changed the text accordingly.

Reviewer 2 Report

Comments and Suggestions for Authors

The manuscript titled Integrin-linked kinase (ILK) promotes mitochondrial dysfunc-2 tion by decreasing CPT1A expression in a folic acid-based 3 model of kidney disease is interesting research, with relevant results presented, however there are few issues:

- Please elaborate how the sample size was calculated, the number of animals used is not clear (e.g. 4-8 animals per group is not specific), and please elaborate why all animals were not used for all analysis.

- page 4, line 114, 115: Please elaborate the methodology of picture analysis.

- page 5, line 130, 131 - Please elaborate further the sentence and methodology: Collagen accumulation in the renal cortex was quantified as red intensity in arbitrary units (a.u.). 

page 14, line 391, 392 - Please elaborate further how the tissues were analyzed.

Author Response

COMMENTS 1

Minor comments: The manuscript titled Integrin-linked kinase (ILK) promotes mitochondrial dysfunc-2 tion by decreasing CPT1A expression in a folic acid-based 3 model of kidney disease is interesting research, with relevant results presented, however there are few issues:

RESPONSE 1

We sincerely appreciate the reviewers´ valuable suggestions for improving our article. Following their recommendations, we have rearranged several sections of the original text. Please, find the changes in red font throughout the entire manuscript. We double checked the English grammar with a bilingual co-worker and changed the text accordingly.

COMMENTS 2

- Please elaborate how the sample size was calculated, the number of animals used is not clear (e.g. 4-8 animals per group is not specific), and please elaborate why all animals were not used for all analysis.

RESPONSE 2

As suggested, we added in each figure legend the group of animals or experiments performed. In the present work we used the FA animal model for studying the pathological mechanisms of AKI to CKD transition, because it is simple and reproducible. However, it is known that FA is a very aggressive treatment in vivo and can reduce the survival of the animals treated.

Although we started from the same number of animals in all groups, FA treated groups increased their mortality. Thus, the number of animals at the end of the experiment decreased in those groups compared with non-FA-treated animals.

Here are two examples of publications where FA-based model was used in vivo and explain the same problem about the reduction in surviving FA-treated mice:

1- Martin-Sanchez, D., Fontecha-Barriuso, M., Carrasco, S., Sanchez-Nino, M. D., Massenhausen, A. V., Linkermann, A., et al. (2018). TWEAK and RIPK1 mediate a second wave of cell death during AKI. Proc. Natl. Acad. Sci. U. S. A. 115, 4182–4187. doi: 10.1073/pnas.1716578115

2- Hamid AK, Pastor Arroyo EM, Calvet C, Hewitson TD, Muscalu ML, Schnitzbauer U, Smith ER, Wagner CA, Egli-Spichtig D. Phosphate Restriction Prevents Metabolic Acidosis and Curbs Rise in FGF23 and Mortality in Murine Folic Acid-Induced AKI. J Am Soc Nephrol. 2024 Mar 1;35(3):261-280. doi: 10.1681/ASN.0000000000000291.

COMMENTS 3

- page 4, line 114, 115: Please elaborate the methodology of picture analysis.

RESPONSE 3

Following the recommendations, we elaborated further the methodology of picture analysis (lines 129 to 137)

COMMENTS 4

- page 5, line 130, 131 - Please elaborate further the sentence and methodology: Collagen accumulation in the renal cortex was quantified as red intensity in arbitrary units (a.u.).

RESPONSE 4

Following the recommendations, we elaborated further the methodology of picture analysis (lines 149 to 159)

COMMENTS 5

page 14, line 391, 392 - Please elaborate further how the tissues were analyzed.

RESPONSE 5

Following the recommendations, we elaborated further the methodology of the procedures used to analyze tissue in the methodology section, subsections 4.3, 4.5, 4.6.  (starting at line 503)

Round 2

Reviewer 1 Report

Comments and Suggestions for Authors

The authors ammended the manuscript according my sugestión.